# Neutrophil N1 and N2 Subsets and Their Possible Association with Periodontitis: A Scoping Review

**DOI:** 10.3390/ijms232012068

**Published:** 2022-10-11

**Authors:** Luis Daniel Sansores-España, Samanta Melgar-Rodríguez, Rolando Vernal, Bertha Arelly Carrillo-Ávila, Víctor Manuel Martínez-Aguilar, Jaime Díaz-Zúñiga

**Affiliations:** 1Faculty of Dentistry, Autonomous University of Yucatán, Merida 97000, Mexico; 2Periodontal Biology Laboratory, Faculty of Dentistry, University of Chile, Santiago 8380492, Chile; 3Department of Medicine, Faculty of Medicine, University of Atacama, Copiapo 7500015, Chile

**Keywords:** phenotype, periodontitis, microbiota, bacteria, host interaction pathogens

## Abstract

Periodontitis is a chronic non-communicable disease caused by dysbiotic changes that affect the subgingival microbiota. During periodontitis, neutrophils play a central role in the initial recognition of bacteria, and their number increases with the appearance of the first signs of periodontal inflammation. Recent evidence has led to the proposition that neutrophils can also functionally polarize, determining selective activity patterns related to different diseases. Two well-defined neutrophil phenotypes have been described, the pro-inflammatory N1 subset and the suppressor N2 subset. To date, it has not been established whether these different neutrophil subtypes play a role in the pathogenesis of periodontitis. Thus, this scoping review aimed to determine whether there was evidence to suggest that the neutrophils present in periodontal tissues can be associated with certain phenotypes. The research question, population, concept, and context sought to identify original articles, in humans, that detected the presence of neutrophils in the periodontal tissues of people affected by periodontitis. Based on the search strategy, we found 3658 studies. After removing the papers with abstracts not related to the outcome measures and eligibility criteria, 16 articles were included for qualitative analysis. Several studies identified the presence of different neutrophil subsets, specifically, the naive, pro- and para-inflammatory, hyper-reactive and hyper-active, and high- and low-responder phenotypes. The existing evidence demonstrates the presence of pro-inflammatory, hyper-reactive and high-responder neutrophils in periodontal tissues affected with periodontitis. There is no evidence demonstrating the presence of the N1 or N2 phenotypes in periodontal tissues during periodontitis. However, the existence of pro-inflammatory phenotypes, which increase NETosis and degranulation, and increase the production of pro-inflammatory cytokines, could be suggestive of the N1 phenotypes.

## 1. Introduction

Neutrophils are polymorphonuclear cells derived from bone marrow granulocyte precursors [1]. Neutrophils have a relatively short lifespan in the peripheral blood circulation, approximately 4 to 5 h. However, a longer lifespan is evidenced when they infiltrate the connective tissues, where they recognize microbiota- or pathogen-associated molecular patterns (MAMPs or PAMPs). Neutrophils are between 60 to 70% of the total polymorphonuclear cells [2]. Neutrophils are the first cell population to be recruited at inflamed sites, and can migrate by chemotaxis from the peripheral blood circulation to the tissues. In the connective tissues, they fulfill their main role: phagocytosis [3,4]. In recent years, in tumor-related pathologies, two subtypes of neutrophils have been defined, with phenotypic and functional differences: neutrophil type 1, or anti-tumorigenic N1, and N2, or pro-tumorigenic neutrophils [5,6]. Additionally, these same phenotypes have been identified as pro-inflammatory (N1) and immune modulatory (N2), based on their abilities to degranulate, release cytokines, and migrate.

The TAN differentiate in the presence of interferon (IFN) type I toward the N1 subset, which is characterized by increasing adhesion, transmigration, phagocytosis, oxidative burst, degranulation, and neutrophil extracellular trap (NET) release, also known as NETosis, as well as cluster of differentiation (CD)-177 over-expression [5]. Otherwise, in the presence of high concentrations of transforming growth factor-β (TGF-β), neutrophils polarize into a modulatory phenotype, fulfilling an antagonistic role to the N1 phenotype, and N2, which also increases the expression of Olfactomedin 4 (OLFM4) [7]. Thus, the phenotypic shift from N1 to N2 may therefore suggest antagonistic signaling pathways between TGF-β and IFN type 1, with a different role during health or cancer [8]. Recently, some experimental studies in rodents have detected a similar distribution of neutrophil subsets in both heart and brain affected with inflammatory diseases.

Additionally, after a heart or brain infarction, neutrophils infiltrate the tissues rapidly after an ischemic event [9]. It was observed that during the first 24 h after ischemia, neutrophil infiltration is essential for the inflammatory response [10,11]. Additionally, it was observed that N2 neutrophils increase over time, and the decrease in the N1/N2 ratio is directly associated with the resolution of inflammation [9]. In fact, in animals depleted of neutrophils and affected by brain ischemia, it was observed that the size of the lesion is smaller compared to animals with neutrophils [12]. In general terms, both in the heart and in the brain after ischemia, the pro-inflammatory milieu triggers the polarization of N1 neutrophils [9,10,11,12,13,14]. The N1 phenotypes increase the pro-inflammatory response by secreting a higher amount of IL-1β, IL-6, TNF-α, and IFN-γ, by producing reactive oxygen species (ROS), and ultimately through NETosis [3,4,15].

Although the TAN1 and TAN2 response has been studied extensively, the possible role of neutrophil phenotypes in other pathologies has only recently been explored in animal models. To date, it has not been identified in chronic inflammatory diseases such as periodontal diseases. Therefore, this scoping review aims to identify whether there is evidence to suggest the presence of these neutrophil subsets in periodontal tissues affected by periodontitis.

## 2. Results

### 2.1. Search Results

Initially, 3658 studies were identified. After removing the papers with abstracts not related to the primary, secondary or tertiary outcome measures, a total of 147 potentially eligible abstracts were selected. When applying the eligibility criteria, 101 articles were selected, and 46 were excluded. After eliminating the duplicates, a full-text analysis of the studies was performed, and, finally, 14 articles were included for qualitative analysis (Figure 1). The studies included in the selection corresponded to four descriptive [16,17,18,19], four cohort [20,21,22,23], and six case–control [24,25,26,27,28,29] studies. The full data summaries are presented in Table 1.

The selected articles correspond to studies in humans characterizing the presence of neutrophils in periodontal tissues based on their functions. Table 1 summarizes the main findings described. In addition, Table 2 shows the main findings for each study, indicating the results that answer our research question and gaps in the research. 

### 2.2. Inclusion of Sources of Evidence

In Table 2, the relevance of each selected source to the objectives of this scoping review is identified.

### 2.3. Review Findings

#### 2.3.1. Phenotypic Characterization of Infiltrating Neutrophils in Periodontitis

During periodontitis, there is an increase in neutrophils in the periodontal tissues compared to the absence of periodontitis [20,26]. Neutrophils are characterized by CD10+, CD11b+, CD15+, CD16+, CD18+, CD55+, CD62L+, CD63+, CD64+, CD66+, CD138+, CD177+, and elastase+ [20,23,26]. In addition, they exhibit increased chemotaxis, phagocytosis, degranulation, production of cytokines, MPO and ROS, and NETosis [27,28,29]. In fact, in periodontitis-affected tissues, NET degradation is lower than in healthy subjects [26]. Additionally, 2.4% of oral neutrophils produce the receptor activator of the nuclear factor-κB ligand (RANKL) [30]. Additionally, in patients with refractory periodontitis, neutrophil ROS production is correlated with increased periodontal index (PI), bleeding on probing (BOP), pocket deep (PD), and clinical attachment level (CAL) [18]. This ROS production is independent of the chemotaxis capability and could be induced by local IFN-γ production [17,25]. 

When analyzing the neutrophil phenotypes described, there is no clear consensus. Some studies mention hyper-reactive or hyperactive phenotypes [16], naïve, pro-inflammatory, and para-inflammatory subsets [21,27]. These phenotypes can be described based on their capability to produce ROS [16,18], their euchromatin/heterochromatin ratio, and their higher or lesser phagocytic activity [21,27]. Interestingly, when patients are treated and a follow-up of 3, 6, or 12 months is carried out, it is observed that the CD16b^+^CD66^+^CD11b^+^CD62L^−^ population decreases, and has a positive correlation with the BOP index [23]. Curiously, CD177^+^ cells predominate in gingival crevicular fluid (GFC) during periodontitis, while the CD177^-^ population predominates under healthy conditions. Curiously, CD177^+^ cells have a higher apoptotic activity than CD177^−^ cells [29].

Finally, when comparing circulating with oral neutrophils, the oral neutrophils present a unique transcriptome, characterized by increased IL-17, IFN-γ, RANKL, CXCL10, and elastase expression [19,30,31].

#### 2.3.2. Para- and Pro-Inflammatory Neutrophils in Periodontitis

Periodontal tissues can be affected by different causes that induce inflammation. Periodontitis has been recognized as the main chronic inflammatory disease, but it is not the only one. There may be a form of periodontitis resulting from trauma, or there may be an adaptation phenomenon [32]. Thus, when periodontitis occurs due to dysbiosis of the microbiota, we speak of pro-inflammatory phenomena, and when periodontitis is due to an adaptive process, we speak of para-inflammatory phenomena [33,34]. This is noteworthy to mention, because the cause of the inflammation defines the type of inflammatory response, and the pro- and para-inflammatory phenotypes have been described in periodontal tissues [21]. In this context, a para-inflammatory and pro-inflammatory phenotype characterized by low granulation, light cytoplasm, and a large amount of euchromatin was detected [21]. Additionally, in healthy periodontal tissues, two different populations of oral neutrophils have been observed—para-inflammatory 1 and para-inflammatory 2—and both of them have a similar profile to naïve blood neutrophils and a lower state of activation [21].

#### 2.3.3. Degranulation and NETosis

Considering degranulation and NETosis as an important part of neutrophil function, it is curious that only one study has evaluated NETosis [16]. The N1 phenotypes produce NET to induce the pro-inflammatory response, and aggNET enhances the N1 function. During periodontitis, the infiltration of neutrophils with high capability of triggering NETosis subsets is essential for determining the greater or lesser recruitment of phagocytes, which influences the increase or decrease in inflammatory signs. Additionally, it is not unusual to find a positive correlation between the presence of NET-producer neutrophils with the PI, and BOP index, or even CAL loss.

## 3. Methods

### 3.1. Review Question

Is there evidence characterizing the existence or the possible role of the N1 and N2 neutrophil phenotypes in periodontal tissues affected by periodontitis?

### 3.2. Participants, Concept, and Context

We include studies conducted in humans, in which the population is clearly identified. In particular, studies were selected where samples of adults between 20 and 80 years of age were used, regardless of the distribution by sex, socioeconomic level or ethnicity (participants). Additionally, the studies described both the periodontal diagnosis and the clinical criteria used to diagnose (concept). We selected the studies that analyzed or characterized the presence of neutrophils in the periodontal tissues of healthy people or those affected by periodontitis (concept), and the origin of each participant (context).

### 3.3. Source of Evidence

The present scoping review was carried out following the indications of the Preferred Reporting Items for Systematic Reviews and Meta-Analyses (PRISMA) guidelines. The protocol related to the data search, selection, extraction, and analysis was discussed and established by all of the authors in a dynamic drive and without further modifications. There were no language, year, or publication status restrictions for inclusion, and when eligible studies in languages other than English, Spanish, French or Portuguese were detected, a qualified translator was consulted. The selected articles were included in a drive for full text analysis, and the articles not selected were included in a table indicating the reasons for their non-inclusion or exclusion.

We included primary research studies, randomized or non-randomized clinical trials, and case–control and cohort studies. We excluded systematic reviews with or without meta-analysis from the analysis, as they are not a primary source of information. However, we left open the option to include any source of evidence directly associated with the research question and that met the eligibility criteria.

### 3.4. Search Strategy

An electronic search was performed in the Latindex, SciELO, Cochrane Library (CENTRAL) and Medline via Pubmed databases, from 30 January 2021 to 30 June 2022. Two reviewers (DS-E and JD-Z) performed the electronic database survey independently and in duplicate. The search strategy was performed using Mesh terms “neutrophil”, “bacteria”, “host–pathogens interactions”, “periodontitis”, “subsets”, and “phenotypes” in the Medline database and adapted for the other databases. The complete search strategy is provided in Appendix A.

### 3.5. Source of Evidence Screening and Selection

Evidence screening and selection were performed independently and in duplicate, by assessing titles and abstracts to determine their inclusion. Furthermore, the selected full-texts were analyzed to determine whether they met the inclusion criteria. In case of disagreement, the article selection was discussed until a consensus was reached on whether the article could be included or excluded. If excluded, the motifs were recorded in the PRISMA flow chart (Figure 1). In cases where consensus was not reached, a third reviewer made the decision (SM-R).

### 3.6. Data Extraction

For each paper included, the following information was extracted: first author and year of publication, study objective, population (number, sex, age), concept (periodontal diagnosis criteria, interventions or phenomena of interest, samples, analysis methods, and results), and context (group definition and origin) (Table 1).

### 3.7. Outcome Measures

To assess the characterization of the neutrophils in the gingival tissues, the primary outcome was to determine the population of neutrophils in the periodontal tissues of patients with or without periodontal disease. As a secondary outcome, articles that isolated neutrophils from peripheral blood samples and evaluated phenotypes were selected. As a tertiary outcome, studies that included a complete phenotype or cytokine analysis in patients with inflammatory diseases were chosen.

## 4. Discussion

Periodontitis is one of the most prevalent pathologies in the oral cavity, which affects the supporting tissues of the teeth: periodontal ligament, root cement and alveolar bone; and its characteristic is the progressive destruction of these tissues with the possibility of a subsequent tooth loss [35,36]. Additionally, it is considered a non-communicable chronic disease, characterized by a high concentration of pro-inflammatory mediators in the affected periodontal tissue, which can spread to the peripheral circulation and induce a chronic low-grade inflammatory phenotype (CLIP) response [36]. The main cause of periodontitis is the dysbiosis of the subgingival microbiota, which begins with an alteration in the amount of bacteria [34,37].

Generally, periodontitis begins with a pro-inflammatory phenomenon, characterized by the presence of bacteria in the gingiva-dental sulcus and their recognition by Langerhans cells pattern recognition receptors (PRR) within the epithelium [38,39,40]. The Langerhans cells secrete IL-8, which will spread into the connective tissue creating a concentration gradient [38,39,40]. Furthermore, neutrophils can recognize the chemoattractant signal through their CD177 and migrate to the thickness of the epithelium [29,39,40]. Once neutrophils reach the site of inflammation and are activated, they facilitate microbial removal through several different processes, including phagocytosis, generation of myeloperoxidase (MPO) and ROS, release of granular content, cytokine production, and NETosis [24,41,42,43]. 

Different studies have evaluated the frequency of resident neutrophils in the periodontal tissues during health or periodontal disease, which vary from 0 to 100% of the leukocyte infiltrate [44,45]. Neutrophils are cells detected with high frequency in oral cavity, saliva, gingival sulcus, and inside pocket epithelium. Recently, 141 cluster of differentiation (CD) from neutrophils were described in healthy and inflammatory periodontal conditions [46]. Of them, CD11b, CD14, CD15, CD16, and CD62L have been used for its phenotypic characterization [20,23,27,46], CD11b/CD18, CD11c/CD18, CD21, and CD35 for phagocytosis evaluation [23,27,47,48], CD63 (primary granules), CD15 and CD66b (secondary granules), CD11b (tertiary granules), and CD13, CD14, CD18 and CD45 (quaternary granules) for degranulation analysis [15,20,23], and CD177 for maturation [49]. Additionally, two markers have been detected in some neutrophils: OLFM4 within granules, and CD177 in plasma membrane [7,50].

Other characteristics that have been observed in neutrophils are the degranulation capacity and death induced by NETosis. Neutrophils produce NET under different conditions, and it is made up of elastase, histones, MPO, and cathepsin-G, among other proteins [43]. In the presence of bacterial invasion, NETosis causes the release of IL-1β in the surrounding neutrophils and increase the production of chemokines such as CXCR2, CXCL8, CXCL1, CXCL3, and CXCL3, which produce a higher infiltrate of neutrophils favoring inflammation by degranulation [51]. Additionally, the neutrophils trigger the aggregation of NET, which initiates the anti-inflammatory process by trapping dead cells, bacteria, or phagocytes, facilitating the macrophages phagocytosis [52]. Interestingly, the NET production depends directly of the C-type lectin receptor Dectin-1, which is a size microorganism sensor [53]. The activation of Dectin-1 activates elastase, and elastase translocates to the nuclei and promotes histone degradation [43]. Once the concentration of neutrophils in the affected site of bacterial invasion exceeds a certain threshold, the NET begins to aggregate, building the aggNET [52]. AggNETs sequester DNA and proteins from neutrophil granules, and degrade pro-inflammatory cytokines through DNA-bound serine proteases. Indeed, as aggNET increases, more control exists over the pro-inflammatory phenomenon [43,52].

Several studies have determined the pro-inflammatory cytokines profile during periodontitis. Today, it has been widely recognized the role of IL-1β, IL-6, IL-8, IL-10, IL-12, IL-35, TNF-α, IFN-γ, and TGF-β in the balance of periodontal health or disease [34,54,55,56,57,58,59]. In this sense, an increase in granulocytes-colony stimulant factor (G-CSF), IL-6, IL-35, and TGF-β could trigger the differentiation towards N2 phenotypes, the increase of IL-12 and IFN-β, the differentiation towards N1 subsets, and the increase of IFN-γ and TNF-α could trigger the N2 to N1 subset conversion [5,60,61]. Recently, it has been proposed that under bacterial infection, the axis N1-M1 (pro-inflammatory macrophages) has a high phagocytic capacity, and the N2-M2 (modulatory macrophages) has a low ability for bacterial clearance [62,63]. Particularly, macrophages can differentiate toward a M1 phenotype that will promote the inflammatory and destructive response of damaged tissue. Otherwise, the M2 differentiate later and play a role in the resolution of inflammation and tissue regeneration [64]. After the first priming, neutrophils produce IL-8 and CXCL1 that allow adhesion, migration and chemoattraction, and represent a positive feedback loop [65,66]. Both subsets of neutrophils express similar levels of co-stimulatory molecules CD80, CD863, CD86, and HLA-DR, and cytokines, participating in antigen presentation [67]. Thus, N1 neutrophils could secrete IL-12, IFN-γ, and TNF-α, and polarize macrophages and dendritic cells (DCs) toward the pro-inflammatory subsets M1 or DC1, respectively. Conversely, against less pathogenic bacteria, neutrophils will produce IL-10 and TGF-β, which could induce M2- and DC2-modulatory subsets [34,57,59]. 

Aboodi and Johnstone described the high- and low-responder phenotypes based on ROS production, corresponding to the N1 and N2 subsets, respectively [18,24]. Additionally, the presence of neutrophils with CD15, CD16, CD62L, and CXCL10 markers corresponds to the N1 subset, and neutrophils with higher levels of CD11b and lower levels of CD62L correspond to the N2 subset [20,23,27]. In the presence of pathogenic mechanisms, neutrophils are primed by TLR2, TLR4, CD14, or even NOD1 [68]. Both subsets produce divergent amounts of CXCR4, which allows neutrophils to migrate to lymph nodes and present antigens to T lymphocytes. N1 neutrophils produce lower levels of CXCR4, while N2 neutrophils produce higher levels of CXCR4 than the N1 subset [69]. In this sense, the pro-inflammatory response triggered by N1 neutrophils could occur in situ after priming, and the modulatory response triggered by N2 probably occurs in the lymph-nodes. Nevertheless, plausible mechanisms for resolvign the gap in the knowledge regarding the capability of N2 to modulate the adaptive response in periodontitis is still unresolved.

Although we can identify that there is evidence in animals that neutrophils N1 and N2 have been described outside of a tumor environment, there are no studies that have characterized them during periodontal disease or in periodontal tissues [9]. A first approach for neutrophil subsets was related to their high or low density. High-density neutrophils (HDN) are Ly6G^HIGH^CD11b^HIGH^, mature, with segmented nuclei, cytotoxic, with a high capacity of migration, phagocytosis, and oxidative burst, non-suppressive, pro-inflammatory, and anti-tumorigenic [70,71,72]. Conversely, low-density neutrophils (LDN) are Ly6G^HIGH^CD11b^LOW^, mature, with segmented nuclei, non-cytotoxic, with a reduced migration, phagocytosis, and ROS production, suppressive, anti-inflammatory, and pro-tumorigenic. Additionally, HDN were named as belonging to a pro-inflammatory or anti-tumor phenotype, or N1, while LDN were denoted as belonging to an anti-inflammatory or pro-tumor phenotype, or N2 [70,72].

HDN subsets are characterized by Toll-like receptor (TLR)-2, TLR4, TLR5, TLR8, CD11b^LOW^CD49^HIGH^CD177^HIGH^ expression and IL-12 production [73]. HDN subsets also produce higher amounts of TNF, CXCL10, and ROS, express higher levels of ICAM-1, CCL3, and CD95, and lower levels of CXC4, vascular endothelial growth factor (VEGF), and IL-8 [5,74,75,76]. Conversely, LDN neutrophils express TLR2, TLR4, TLR7, TLR9, and CD11b^HIGH^CD49^LOW^CD177^LOW^, and produce IL-10 [73]. LDN has a long lifespan and produces large amounts of arginase, which in turn inactivates T-cells, produce higher levels of CCL2, CXCR2, CXCR4, VEGF, IL-8, IL-10, and TGF-β1, and produce low levels of ICAM-1 and CCL3 [5,75,77]. LDN neutrophils also express MPO and produce ROS, but at a lower rate than HDN [76]. The greater or lesser extent of ROS production can be explained by its pro-inflammatory or homeostatic role. In fact, Aboodi [18] demonstrated the presence of two responder phenotypes, according to ROS production. HDN cells produce an oxidative burst, which contributes to local degranulation and bacterial killing, while LDN decreases ROS in order to produce genotoxicity in other immune cells [76]. When comparing published data, it is possible to identify the presence of different neutrophil phenotypes; however, whether they are N1 or N2 subsets can only be speculated (Figure 2). Curiously, in systemic inflammation, a third phenotype has been described, characterized by being CD62^LOW^CD11b^HIGH^CD11c^HIGH^, and by its strong capacity to inhibit T lymphocyte phenotypes by direct contact through Mac1 integrin and ROS production [53,78]. Additionally, two studies analyzed a population of neutrophils, detecting a third circulating phenotype, which could be the granulocytic myeloid-derived suppressor cells (gMSCD) [16,23,72].

In general terms, there are no human studies demonstrating the presence of the N1 or N2 phenotypes of neutrophils in periodontal tissues. The closest approximations are related to the increase in pro-inflammatory functions in people with periodontitis, which in itself does not imply that there is an N1 phenotype. Thus, the absence of evidence does not allow us to demonstrate that these phenotypes exist in the periodontium, nor does it allow us to identify a possible role. Thus, in vitro and experimental studies are required to demonstrate the role of the two neutrophil phenotypes during periodontitis.

## 5. Future Research Directions

The evidence of neutrophil subsets in periodontitis is an unexplored field. Even though several studies have determined different neutrophil roles, the role of the N1 or N2 subsets in periodontal health or disease still cannot be concluded. It is necessary to determine the pathways by which oral bacteria may be able to induce N1 or N2 response in in vitro or experimental models. Additionally, it is necessary to determine the frequency of detection of these neutrophil phenotypes in periodontal health or disease.

## 6. Conclusions

The presence of neutrophils increases in tissues affected by periodontitis compared to healthy tissues. These neutrophils have a pro-inflammatory phenotype characterized by increased phagocytosis, degranulation, production of pro-inflammatory cytokines, and NETosis. However, neither study defines detected neutrophils as being either N1 or N2.

## 7. Implication of the Findings for Research

This scoping review demonstrates that there is a gap in knowledge. In this context, new research hypotheses can be developed that must be resolved in the future. Additionally, Figure 3 represents a hypothesis regarding the role of both subsets during periodontitis, which must be proven in the future.

## 8. Implication of the Findings for Practice

Understanding the role of neutrophils as the first line of immune response will make it possible to design therapeutic alternatives for the treatment of pro-inflammatory diseases in order to avoid N1 polarization or induce N2 polarization. Specifically, periodontitis is caused by keystone pathogens that trigger a pro-inflammatory response. In this context, determining what type of neutrophil response each bacterium induces would be important for assessing the microbiological or immunological susceptibility of each individual.

## Figures and Tables

**Figure 1 ijms-23-12068-f001:**
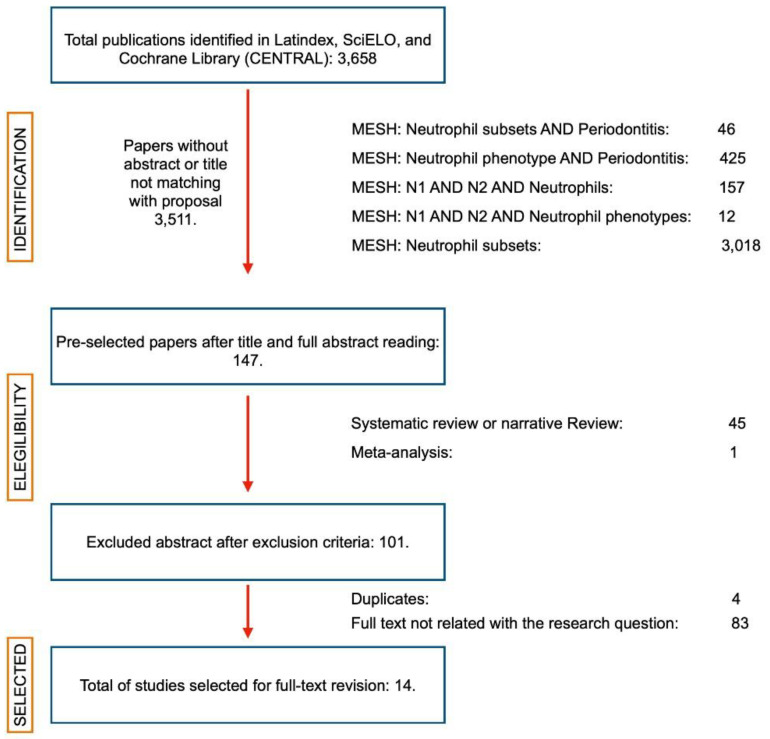
PRISMA flow diagram. Summary of the article search. From a universe of 3658 articles, 147 passed the first review of abstract and title. Subsequently, systematic reviews with or without meta-analysis and narrative reviews were eliminated until 101 articles remained, which were reviewed in full-text format. Finally, 14 articles were selected to prepare this systematic review. (Created with http://biorender.com, accessed on March 2020).

**Figure 2 ijms-23-12068-f002:**
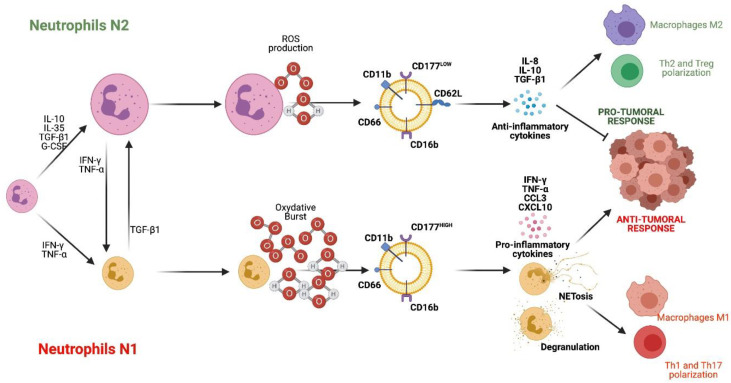
Neutrophil subsets N1 and N2. N2 neutrophils are differentiated in the presence of IL-10, IL-35, TGF-β1, and G-CSF. N2 cells are characterized by producing ROS, IL-10 and TGF-β1, expressing CD11b, CD16b, CD62L, CD66 and CD177LOW, and having a pro-tumor and anti-inflammatory function. On the other hand, N1 neutrophils differentiate in the presence of IFN-γ and TNF-α. They are characterized by producing oxidative burst, IFN-γ, TNF-α, CCL3, CXCL4 and expressing CD16b, CD11b, CD66 and CD177HIGH, and fulfill an anti-tumor and pro-inflammatory role. References: [16,18,23,24,72,76]. (Created with http://biorender.com).

**Figure 3 ijms-23-12068-f003:**
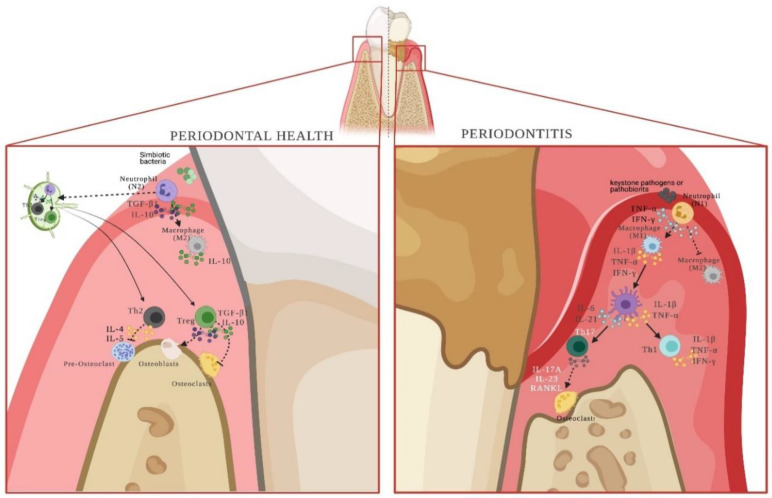
N1 and N2 neutrophils and their role during periodontal health or disease. Neutrophils, by recognizing symbiotic bacteria, have the ability to differentiate into the N2 phenotype. Cytokines produced by N2 cells will polarize macrophages towards M2-anti-inflammatory subsets. In addition, they will migrate to the regional lymph-node and present the antigen to naïve lymphocytes to allow their differentiation into the Th2 and Treg phenotypes, which will naturally maintain tissue homeostasis. Conversely, in the presence of keystone bacteria or pathobionts, neutrophils polarize towards the N1 phenotype, which secretes cytokines that allow the differentiation of pro-inflammatory M1 macrophages and pro-inflammatory dendritic cells. Together, they present antigens to T lymphocytes at the affected periodontal site, differentiating them towards Th1 and Th17 effector phenotypes. Thus, the response initiated by N2 neutrophils is a regional response, and the one triggered by N1 neutrophils is localized. References: [5,34,57,59,60,61,62,63,64,65,66,69]. (Created with http://biorender.com).

**Table 1 ijms-23-12068-t001:** Data extracted after individual analysis. Population, concept, and context individualization.

.	Author/Year	Objective	Participants				Concept				Context
			Total Number	Groups	Sex (Female %)	Age	Periodontal Diagnosis Criteria	Interventions or Phenomena of Interest	Samples	Results	Origin
1	Aboodi et al., 2011 [18]	To identify the presence of oral neutrophil hyperactivity among refractory periodontitis patients, and to determine if the hyperactivity is related to a history of periodontal disease severity.	13	No	46.15	32 to 73 years	Not specified	Periodontal screening	Venous blood samples	Characterization of high-responder neutrophils in patients with periodontitis, and low-responder neutrophils in healthy subjects.	Refractory Disease Unit, Dental Research Institute, University of Toronto, Canada.
									Oral rinse sample		
2	Borenstein et al., 2018 [21]	To determine the morphological diversity between different groups of patients, as well as the response induced in naïve neutrophils after incubation with bacteria.	5	Health	40	46.1 ± 17.9	Not specified	None	Blood and oral expectorated samples	Oral neutrophils had lower number of granules per cytoplasm area compared to blood neutrophils. Bacteria-stimulated oral neutrophils were more granular, had more phagosomes than the blood neutrophils. Neutrophils that migrated into the connective tissue have a lighter cytoplasmic density, fewer granules and higher euchromatin fraction. Gingival tissue neutrophils had increased euchromatin/heterochromatin ratio.	Toronto General Hospital 2 Nephrology Center and University of Toronto’s Graduate Periodontology Clinic, Toronto, Canada.
			6	Diseased	50	58.0 ± 15.1					
3	Dutzan et al., 2016 [20]	To characterize the human immunological cell network patrolling the oral barrier in health with a particular focus on the gingival area.	50	Health	70	26.5	PPD, CAL, BOP and dental radiographs	None	Biopsies	An increase was observed in the proportion of CD15+CD16+ neutrophils in gingival biopsies. Additionally, CD15+CD16+ cells increased in gingival tissues affected with periodontitis.	NIH Clinical Center
			6	Diseased	66.67	38.5					
4	Fine et al., 2016 [27]	To demonstrate the presence of para-inflammatory neutrophils in the healthy oral cavity and pro-inflammatory neutrophils in patients with periodontitis.	11	Health	54	26 to 84	PPD, CAL, BOP, and recession	None	Blood and oral samples	Oral neutrophils from patients with periodontitis are in a pro-inflammatory activation state when compared with healthy oral neutrophils. Neutrophils from patients affected with periodontitis were characterized by elevated degranulation, phagocytosis, ROS production, and NETosis.	Toronto General Hospital’s Nephrology Center and the University of Toronto’s Graduate Periodontology Clinic
			17	Diseased	52	26 to 82					
5	Johnstone et al., 2007 [24]	To compare the generation of oxygen radicals in peripheral neutrophils from patients with aggressive, chronic, and periodontally healthy after stimulation with phorbol myristate acetate (PMA). Additionally, to examine the phagocytotic ability of the neutrophils.	22	Diseased	59%	45.83	CAL loss despite adequate maintenance therapy	None	Blood samples	Activated neutrophils demonstrated hyperactive NADPH oxidase activity in patients with refractory aggressive periodontitis compared to patients successfully treated and healthy.	Graduate Periodontal Clinic, Faculty of Dentistry, University of Toronto
			13	Healthy	46.15%	33.25					
6	Lakschevitz et al., 2013 [19]	To present a rapid method for oral neutrophil isolation and to characterize and compare the neutrophil gene expression profile in the blood and oral compartment of healthy individuals.	5	Healthy	40%	23 to 38	PPD, OP, PI.	None	Blood samples	Oral neutrophils expressed TCRs. Neutrophils presented a site-specific gene expression profile in the oral cavity of healthy individuals when compared with blood neutrophils.	Faculty of Dentistry, University of Toronto
7	Matthews et al., 2006 [16]	To confirm the reported FcgR hyper-reactivity of peripheral neutrophils in chronic periodontitis using more relevant physiological conditions, and to determine whether the ROS could be detected.	18	Diseased	72.2%	47.2% +/− 6.1	PPD, BOP, CAL, Radiography	None	Blood sample	Peripheral neutrophils from patients with chronic periodontitis exhibited both hyper-reactivity and hyperactivity.	Birmingham Dental Hospital
8	Medara et al., 2020 [23]	To assess the longitudinal variation in the expression of the adhesion and activation markers of neutrophils, to determine the neutrophil maturation stage based on CD surface markers, and to evaluate the suppressive neutrophil phenotypes.	40	Health	75%	49.3% +/− 10.6	PPD, BOP	Non-surgical periodontal treatment.	Blood sample	PD and BOP correlated with neutrophil subsets. In particular, neutrophils over-expressed CD11b, CD16b, and CD66b, and under-expressed CD62L.	The Royal Dental Hospital of Melbourne and Melbourne Dental Clinic, The University of Melbourne.
			54	Diseased	63%	53.28% +/− 11.4					
9	Papantonopoulos et al., 2019 [22]	To investigate in datasets of immunologic parameters from early onset and late-onset periodontitis patients (EOP and LOP), the existence of hidden random fluctuations (anomalies or noise), which may be the source for increased frequencies and longer periods of exacerbation, resulting in rapid progression in EOP.	18	Early onset localized periodontitis	50%	19.9 +/− 6.5	Not specified	None	Raw data set	In early-onset periodontitis, there was an increase in IL-1, IL-4, and IFN-γ compared with late-onset periodontitis. CD8, CD20, CD4/CD8 ratio and IL-2 were higher in late-onset periodontitis. Additionally, chemotaxis, phagocytosis, and adhesion of neutrophils were higher in late-onset periodontitis, than early-onset.	Okayama University Dental Hospital
			50	Early onset generalized periodontitis	43.2%	28.3 +/− 5.8					
			43	Late-onset periodontitis	65.4%	47.0 +/− 11.0					
10	Rudin et al., 2020 [29]	To determine whether the CD177+ and CD177− neutrophil subsets differ in their propensity to migrate to both aseptic- and microbe-triggered inflamed human tissues.	Not specified						Buffy coats	CD177+ neutrophil subtype was recruited to inflammatory exudate in periodontitis. Increased levels of CD177+ neutrophils in blood of periodontitis patients were detected, as compared to healthy controls.	Specialist Clinic of Periodontics in Gothenburg, Public Dental Health Services, Region Västra Götaland, Sweden
11	Thorbert-Mros et al., 2014 [26]	To analyze differences in cell characteristics between lesions representing long- standing gingivitis and severe periodontitis	28	Gingivitis	50%	59.5% +/− 7.9	PPD, CAL, BOP, and recession	None	Periodontal tissues biopsies	In periodontitis lesions, there existed an increase in CD138+ and elastase+ cells compared with gingivitis. Elastase+ cells were detected in proximity to the pocket epithelium.	Clinics of Periodontics in Gothenburg and Mölndal, the Clinic for Undergraduate Training in Gothenburg, Public Dental Health Services, Region Västra Götaland and Institute of Odontology
			36	Periodontitis	47%	52.3% +/− 9.5					
12	White et al., 2016 [28]	To investigate ex vivo peripheral neutrophil extracellular trap (NET) production and their subsequent degradation by plasma in chronic periodontitis patients, and periodontally and systemically healthy-matched controls.	40	Healthy UK	46 +/− 8	Not indicated	PPD, CAL, BOP, GI, PI.	Prevention	Blood samples	NET degradation was lower in patients with periodontitis compared with controls.	Specialist Periodontal Centres in Birmingham, UK and Thessaloniki, Greece.
				Healthy Greece	50 +/− 11						
			40	Periodontitis UK	46 +/− 8			Periodontal treatment			
				Periodontitis Greece	52 +/− 8						
13	Wright et al., 2008 [16]	To use microarray technology to analyze the gene expression signature of hyperresponsive PBN from periodontitis patients to identify factors potentially important for disease pathogenesis.	19	Healthy	46.4 +/− 5.4	73.6%	Previously, reported	None	Blood samples	In patients with periodontitis, there was an increase in the IFN-stimulated genes associated with neutrophils. Additionally, IFN-γ prime neutrophils for ROS generation.	Birmingham Dental Hospital (Birmingham, U.K.)
			19	Periodontitis	47.2 +/− 6.1	73.6%					
14	Yagi et al., 2009 [17]	To test the hypothesis that PDK1 regulates chemotaxis in neutrophils and is responsible for the abnormal neutrophil chemotaxis in LAP.	Not specified					None	Blood samples	PDK1 is an absolute requirement for neutrophil chemotaxis. PDK1 is not involved in superoxide production. Thus, inhibition of PDK1 blocked neutrophil migration but not Superoxide production.	Boston University Institutional Review Board, Boston, USA.

PPD: probing of pocket depth, CAL: clinical attachment loss, BOP: bleeding on probing, OP: oral plaque, PI: plaque index, GI: Gingival index, CD: cluster of differentiation, NET: neutrophil extracellular trap.

**Table 2 ijms-23-12068-t002:** Contribution of the selected studies to the outcome.

		Primary Outcome	Secondary Outcome	Tertiary Outcome
	Author/Year	*To determine the population of neutrophils in the periodontal tissues of patients with or without periodontal disease*	*To isolate neutrophils from peripheral blood samples and evaluate phenotypes.*	*To determine the neutrophils phenotype or cytokine analysis in patients with inflammatory diseases*
1	Aboodi et al., 2011 [18]			
2	Borenstein et al., 2018 [21]			
3	Dutzan et al., 2016 [20]			
4	Fine et al., 2016 [27]			
5	Johnstone et al., 2007 [24]			
6	Lakschevitz et al., 2013 [19]			
7	Matthews et al., 2006 [16]			
8	Medara et al., 2020 [23]			
9	Papantonopoulos et al., 2019 [22]			
10	Rudin et al., 2020 [29]			
11	Thorbert-Mros et al., 2014. [26]			
12	White et al., 2016 [28]			
13	Wright et al., 2008 [16]			
14	Yagi et al., 2009 [17]			

## Data Availability

The study did not report any data.

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
