# Peer review of "Neutrophil N1 and N2 Subsets and Their Possible Association with Periodontitis: A Scoping Review"

_ijms, 2022, doi:10.3390/ijms232012068_

Round 1
Reviewer 1 Report
The 'scoping review' presented the hypothetical role of two phenotypes of neutrophils N1 and N2 in onset and progression of periodontitis.
Although, the primary concept of the study appears novel and intriguing, there has been an obvious lack of structured critical approach during literature 'scoping' review. Such review requires both robust content to verify a discussed hypothesis and potential aetiological factors.
Critical evaluation and discussion highly recommended, with evidence-based data. Presentation of limitations of available studies and their results.
A comprehensive table comparing current scientific data is essential.
Literature review needs a substantial re-arrangement, with more logical sequence.
Explanations required whether authors have enough expertise to review and discuss specific aspects of cells interactions. Did they consult their ideas with specialists in physiology/pathology/histology and other relevant topics?
The conclusions are confusing, a clarification needed.
Minor comments:
References must meet MDPI criteria.
Some paragraphs need rephrasing to avoid notorious repetition.
Author Response
The 'scoping review' presented the hypothetical role of two phenotypes of neutrophils N1 and N2 in onset and progression of periodontitis.
Although, the primary concept of the study appears novel and intriguing, there has been an obvious lack of structured critical approach during literature 'scoping' review. Such review requires both robust content to verify a discussed hypothesis and potential aetiological factors.
R: We appreciate the critical analysis and make a modification to the form and content of the manuscript. We apologize for not having structured it as a Scoping Review from the beginning. Therefore, we modified the introduction, making it more conducive to the objective. We include the methodology used, as well as a new Fig of the PRISMA analysis (Fig. 1) and the table with the information collected (Table 1).
Critical evaluation and discussion highly recommended, with evidence-based data. Presentation of limitations of available studies and their results.
R: We include the discussion item, where we carry out the critical analysis of the evidence, with the respective analysis of the limitations of the current studies.
A comprehensive table comparing current scientific data is essential.
R: We include a table that summarize the scientific data (table Nº1).
Literature review needs a substantial re-arrangement, with more logical sequence.
R: we re-arrange the manuscript in order to have a logical sequence.
Explanations required whether authors have enough expertise to review and discuss specific aspects of cells interactions. Did they consult their ideas with specialists in physiology/pathology/histology and other relevant topics?
R: authors have expertise in all the aforementioned topics, thus we improve the results and discussion.
The conclusions are confusing, a clarification needed.
R: changes were made.
Minor comments:
References must meet MDPI criteria.
R: references were changed to MDPI criteria.
Some paragraphs need rephrasing to avoid notorious repetition.
R: We appreciate this review, indeed, there were duplicate phrases or content, which were reordered.
Reviewer 2 Report
The authors have described the neutrophil role in periodontitis and directed the importance of neutrophil subset characterization in the pathology. The authors have described the current understanding very well and the review is of significant interest to the field.
The authors may include the existing macrophage subset studies in periodontitis to highlight the importance unearthing the neutrophil phenotype in the pathology. This strengthens the case and enhance the quality of the review before publishing.
Also, a few grammatical/spelling corrections may be necessary.
Author Response
The authors have described the neutrophil role in periodontitis and directed the importance of neutrophil subset characterization in the pathology. The authors have described the current understanding very well and the review is of significant interest to the field.
The authors may include the existing macrophage subset studies in periodontitis to highlight the importance unearthing the neutrophil phenotype in the pathology. This strengthens the case and enhance the quality of the review before publishing.
R: We appreciate the comments and review made. Also, we include this studies in the discussion.
Also, a few grammatical/spelling corrections may be necessary.
R: the text was modify according the reviewers suggestions, thus, grammatical/spelling were deleted or modified.
Reviewer 3 Report
Dear authors,
Although I deduct that this manuscript is a scoping, you did not follow ths guidelines for scoping reviews (According to the JBI Reviewer’s Manual).
There is a JBI MANUAL FOR EVIDENCE SYNTHESIS: SCOPING REVIEWS CHAPTER and a TEMPLATE FOR SCOPING REVIEW PROTOCOLS which should be followed.
This manuscript, however, looks like an extensive literature search, with no personal contribution to the field of research.
It describes
1. Neutrophils N1 and N2 subsets in tumoral diseases
2. Periodontitis and the potential role of neutrophil subsets
3. Neutrophils N1 and N2 subsets in cardiovascular diseases
4. N1 and N2 neutrophils in periodontitis
Future research directions are not related to the content of the manuscript.
Conclusions are not sustained by the research.
Author Response
Dear authors,
Although I deduct that this manuscript is a scoping, you did not follow the guidelines for scoping reviews (According to the JBI Reviewer’s Manual).
There is a JBI MANUAL FOR EVIDENCE SYNTHESIS: SCOPING REVIEWS CHAPTER and a TEMPLATE FOR SCOPING REVIEW PROTOCOLS which should be followed.
R: we really appreciate the review and suggesting, which undoubtedly will improve the quality of our manuscript. We made several changes in the manuscript, including the introduction, methods, results, and discussion, rearranging it as a Scoping review.
This manuscript, however, looks like an extensive literature search, with no personal contribution to the field of research.
It describes
1. Neutrophils N1 and N2 subsets in tumoral diseases
2. Periodontitis and the potential role of neutrophil subsets
3. Neutrophils N1 and N2 subsets in cardiovascular diseases
4. N1 and N2 neutrophils in periodontitis
R: The mentioned subtitles were removed. Indeed, the contents of subtitles 1 and 2 were incorporated into the introduction section, modifying and directing the content towards the general objective. We include the Methodology section, where we incorporate the figure of the Prisma analysis (Fig Nº1). In addition, in the results section we include a summary table of the selected articles. Finally, in the discussion section, we add new antecedents.
Future research directions are not related to the content of the manuscript.
R: we modified it.
Conclusions are not sustained by the research.
R: we modified it.
Round 2
Reviewer 1 Report
The manuscript's content has been significantly improved compared to original version. Unfortunately, the potentially serious concern occurs, associated with enclosed Figures, specifically copyrights issues.
Several, similar/identical elements present in Figure 1, 2 (cells, femur bone on the left) and 3 (tooth icon on the top, alveolar structure cross-section, cells) can be also found in numerous, previously published articles, describing the role of neutrophils in periodontitis and other inflammatory diseases.
The examples and links are provided below:
https://www.mdpi.com/1422-0067/21/11/3829/htm
https://www.frontiersin.org/articles/10.3389/fimmu.2021.768479/full
https://www.intechopen.com/chapters/67314
https://www.mdpi.com/1422-0067/23/10/5540/htm
https://www.oralhealthgroup.com/features/the-neutrophil-in-oral-health-and-disease-the-new-diagnostic-biomarker/
https://www.cell.com/trends/immunology/fulltext/S1471-4906%2822%2900091-6 (picture of femur bone, Figure 1)
The lack of robust and genuine explanations related to copyrights issues (figures, graphics), applicable for this manuscript, should prevent this article being considered for publication due to the substantial scientific integrity concern.
Author Response
We really appreciate the comments made by the reviewer. We have reviewed the figures of the articles mentioned and in this regard we can inform that our Figures were made on the BioRender online platform. We have used this platform in various articles:
doi: 10.1155/2021/5573937
doi: 10.3389/fragi.2021.781582
doi: 10.1111/odi.14054
Also, the cells and the femur are similar because they were made under the same program. In relation to obtaining the information to design our figures, we must inform that they were designed and made 100% based on the reviewed articles. For this, we have incorporated in the Legend figure the references that were used for its design and the name of the program.
If the reviewer prefers, we might consider deleting Fig. 2 if necessary.
Reviewer 3 Report
Dear authors,
Congratulations on your work!
Author Response
We really appreciate the compliments and comments made by the reviewerRound 3
Reviewer 1 Report
Thank you for providing thorough explanations related to added figures and copyright issue. The extra additions for figures legents, with the appropriate references, support the scientific integrity of these Figures to certain extend.
If the graphics and the modified versions of figures created by BioRender online platform does not require copyrights permission (especially the original graphics available via BioRender), this content is deemed acceptable. The authors needs to ensure that all figures are a genuinely modified versions of stock graphics from BioRender, not exact copies. Otherwise, they must be either removed of appropriately edited in order to create a new, individual version.
The removal of Figure 2 as suggested seems a very reasonable decision.
Author Response
We really appreciate the feedback and have incorporate the suggestion to delete Figure 2. In this way, Figure 3 is now #2 and Figure 4 is now #3.